# Investigating Potential Cardiovascular Toxicity of Two Anti-Leukemia Drugs of Asciminib and Ponatinib in Zebrafish Embryos

**DOI:** 10.3390/ijms231911711

**Published:** 2022-10-03

**Authors:** Huan-Chau Lin, Ferry Saputra, Gilbert Audira, Yu-Heng Lai, Marri Jmelou M. Roldan, Honeymae C. Alos, Charlaine A. Aventurado, Ross D. Vasquez, Guan-Jhe Tsai, Ken-Hong Lim, Chung-Der Hsiao

**Affiliations:** 1Division of Hematology and Oncology, Department of Internal Medicine, Mackay Memorial Hospital, No. 92, Section 2, Zhongshan North Road, Taipei 10449, Taiwan; 2Laboratory of Good Clinical Research Center, Department of Medical Research, Mackay Memorial Hospital, No. 45, Minsheng Road, Tamsui District, New Taipei City 25160, Taiwan; 3Department of Chemistry, Chung Yuan Christian University, Chung-Li, Taoyuan City 320314, Taiwan; 4Department of Bioscience Technology, Chung Yuan Christian University, Chung-Li, Taoyuan City 320314, Taiwan; 5Department of Chemistry, Chinese Culture University, Taipei 11114, Taiwan; 6The Graduate School, Faculty of Pharmacy, University of Santo Tomas, Manila 1008, Philippines; 7The Graduate School, University of Santo Tomas, Manila 1008, Philippines; 8Department of Pharmacy, Research Center for Natural and Applied Sciences, University of Santo Tomas, Manila 1008, Philippines; 9Department of Medicine, MacKay Medical College, New Taipei City 252, Taiwan; 10Center of Nanotechnology, Chung Yuan Christian University, Chung-Li, Taoyuan City 320314, Taiwan; 11Research Center for Aquatic Toxicology and Pharmacology, Chung Yuan Christian University, Chung-Li, Taoyuan City 320314, Taiwan

**Keywords:** zebrafish, anti-cancer drug, asciminib, ponatinib, cardiovascular toxicity, molecular docking

## Abstract

BCR-ABL, a fusion protein kinase, is a druggable target exclusively expressed in patients with chronic myeloid leukemia (CML). Several anti-leukemia medicines targeting this protein have been developed in recent years. However, therapeutic options are limited for CML patients bearing multiple BCR-ABL1 mutations. Ponatinib (PON), a potent tyrosinase inhibitor, was one of the approved drugs for managing BCR-ABL1 T315I mutant disease. However, treatment of patients with PON reported severe side effects related to cardiovascular events. Asciminib (ASC) was the first allosteric inhibitor approved to target the myristoyl pocket of BCR-ABL protein to inhibit protein activity. The different mechanism of inhibition opens the possibility of co-exposure with both medicines. Reports on cardiovascular side effects due to the combination use of PON + ASC in pre-clinical and clinical studies are minimal. Thus, this study aimed to observe the potential cardiovascular-related side effect after co-exposure to ASC and PON using zebrafish as an animal model. In this study, zebrafish were acutely exposed to both compounds. The cardiovascular physiology parameters and gene expression related to cardiovascular development were evaluated. We demonstrate that combining ASC with PON at no observed effect concentration (NOEC) did not cause any significant change in the cardiac performance parameter in zebrafish. However, a significant increase in *nkx2.5* expression level and a substantial decrease in blood flow velocity were recorded, suggesting that combining these compounds at NOEC can cause mild cardiovascular-related side effects.

## 1. Introduction

The breakpoint cluster region-Abelson (BCR-ABL), a fusion oncoprotein with tyrosine kinase activity, is frequently detected in patients with chronic myeloid leukemia (CML). This oncoprotein is derived when part of chromosome 9 is translocated with part of chromosome 22, making a short chromosome 22 called the Philadelphia chromosome [1,2,3]. The Philadelphia chromosome is detected in more than 90% of patients with CML. BCR-ABL oncoprotein allows leukemic cells to avoid apoptosis and increase cell proliferation [4,5]. Targeting BCR-ABL with selective ABL tyrosine kinase inhibitors (TKIs) has revolutionized the treatment of CML and has resulted in an excellent prognosis in patients with chronic phase CML [6]. Imatinib, a selective BCR-ABL TKI, is the first drug explicitly targeting BCR-ABL protein and has been used as the first-line therapy for patients with CML [7]. However, several researchers found that some BCR-ABL proteins resist it not long after its introduction for clinical use. This resistance was frequently related to an acquired mutation found in the ABL kinase domain that impaired the therapeutic effect of imatinib. This incidence led to the development of the second generation BCR-ABL TKI, e.g., nilotinib and dasatinib, that can overcome some imatinib-resistant *ABL1* mutations [8,9]. However, the most important *ABL1* T315I mutation is resistant to first and second-generation TKI.

Ponatinib (PON) is a third-generation BCR-ABL TKI approved by the United States Food and Drug Administration in December 2012 as the second-line treatment of CML and Philadelphia chromosome-positive acute lymphoblastic leukemia (ALL). Unlike the previous generation TKIs, PON can inhibit varieties of BCR-ABL mutants, including the T315I mutant [10]. Furthermore, PON has also been shown to have better efficiency in blocking many tyrosine kinase pathways than other drugs in this category [11]. However, the cardiovascular side effects of PON, including myocardial infarction, arterial thrombosis, and other peripheral-vascular-related diseases, become significant concerns during the clinical development of this drug [12,13,14]. Another newly developed drug for patients with CML is asciminib (ASC). ASC is the first-in-class allosteric inhibitor specifically targeting the ABL myristoyl pocket (STAMP) that works by inhibiting the myristoyl side present in BCR-ABL protein which is distinct from the ATP-binding site of the kinase [15]. As a myristoyl site inhibitor, ASC has the potential to overcome the resistance that happened in other TKIs and open the possibility for dual inhibition with other TKIs that target the ATP-binding site [16,17]. ASC has been reported to cause milder side effects only and is perceived to be generally safe, making ASC a promising drug for CML [18]. Although only reported in a few cases, adverse cardiovascular events such as thrombosis are still reported for ASC use in CML patients. Thus, exploring potential cardiovascular adverse effects of ASC alone or in combination with other TKIs is necessary to rule out its potency in treating CML and reducing cardiotoxicity associated with other TKIs. To our knowledge, only a few studies have focused on the evaluation of the cardiovascular safety profiles of ASC.

Zebrafish is an aquatic vertebrate animal model to study aqua toxicity as many studies have confirmed the similarity of toxic profiles in zebrafish and mammalian models. Previously, zebrafish has been used in many kinds of study including tissue regeneration [19,20], neuroscience [21,22], genetic manipulation [23,24], drug testing in pharmacology, and water pollutant screening [25,26,27,28]. Easy to maintain with a high reproduction rate and sensitivity to variations in water quality make it an excellent animal model for high throughput drug testing. Moreover, 70% of the zebrafish genome was homologous to its human counterpart, making it a good model for genetic studies [29]. At larval stages, zebrafish have a transparent body that makes it possible to observe the development of the cardiovascular system in zebrafish [30]. Zebrafish also have a heart-pumping mechanism almost identical to humans at the molecular level. A previous study has shown that more than 95% of drugs that induce prolonged QT intervals in humans also have a similar effect in zebrafish. All these traits make zebrafish an excellent animal model for studying cardiovascular disease [31].

Using zebrafish as an animal model, we examined the potential cardiovascular effects caused by PON, ASC, and PON + ASC in zebrafish larvae. Several cardiovascular physiology parameters like heart rate, ejection fraction, fraction shortening, stroke volume, cardiac output, and heartbeat regularity were evaluated. The effects on the expression of several biomarkers related to cardiovascular development were also documented to unveil the potential mechanism of PON and ASC in inducing cardiovascular toxicity.

## 2. Results

### 2.1. Overview of the Experiment Design

Before the cardiovascular toxicity assay, we first performed the 96 h acute toxicity assay in the zebrafish model based on OECD 203 guidelines. After the 96 h acute toxicity test, the LC_50_ for PON was estimated as 2.713 ppm. The ASC is less toxic than PON since no apparent death was detected even when the ASC concentration reached 50 ppm. Once the LC_50_ value had been determined, we tested potential cardiotoxicity by incubating zebrafish embryos aged 48 h post-fertilization (hpf) with targeted chemicals (PON, ASC, and PON + ASC) for 24 h. The corresponding cardiac physiology alterations were measured at 72 hpf. Several parameters were measured for cardiac physiology assessment: heartbeat rate, cardiac output, stroke volume, fraction shortening, ejection fraction, and heartbeat regularity. Three important parameters: maximum blood flow velocity, average blood flow velocity, and blood flow signal oscillation pattern, were measured and compared for vascular physiology. After these assays, zebrafish embryos were subjected to a qPCR assay for biomarker expression and mechanism evaluation. The major biomarkers tested in this study were *nkx2.5*, *tbx5*, *myh6*, *amhc*, and *vmhc* for heart development and function; *hbbe1*, *hbbe2*, *hbae1*, and *gata1* for red blood cell oxygen binding and development; *fli1*, *vegfaa*, and *vegfab* for vascular endothelial development (experimental workflow is summarized in Figure 1).

### 2.2. Comparison of Zebrafish Cardiac Performance at Single Chemical Exposure

For pre-testing, we measured the effect of acute 24 h of PON and ASC incubation on the cardiac performance of zebrafish larvae at 72 hpf. The concentrations of PON (0.5, 2.5, 5, and 10 ppm) were selected to investigate the potential acute cardiovascular toxicity. After incubation in 2.5 ppm of PON, we observed that most of the fish underwent cardiovascular-related problems like arrhythmia, cardiac edema, backflow of blood, and some sign of damaged cardiac muscle (Figure 2A–C). Furthermore, every cardiac physiology parameter tested, e.g., stroke volume, cardiac output, shortening fraction, and ejection fraction, also significantly decreased at 2.5 ppm, which shows that at this concentration, the heart pumping capability already decreased significantly (Figure 2D–G). At 2.5 ppm, the cardiac rhythm was also significantly disturbed, as shown by the difference in the heartbeat count at the atrium and ventricle chamber (Figure 2H,I) and the significant increase in the heart rate variability (Figure 2J–M). No data was shown for 5 and 10 ppm as the fish died in this concentration.

Contrary to the PON, acute exposure in ASC did not significantly alter the cardiac performance in zebrafish (Figure 3D–M) even after incubation at 10 ppm. No heart malformation and cardiovascular-related adverse effects were observed after 10 ppm exposure which means that ASC was safer compared to PON in case of a cardiovascular-related event (Figure 3A–C). In the pretesting, no concentration higher than 10 ppm was used for ASC. This was done to prevent false positive results caused by the high concentration of DMSO, as a concentration of more than 1 ppm of DMSO can reduce the heart rate in zebrafish embryos [32].

### 2.3. Zebrafish Cardiac Performance after Co-Incubation in PON and ASC

Based on the results of the single exposure experiment, the concentration of 0.5 ppm of PON and 10 ppm of ASC was selected and further explored to check the potential effect of the combination (PON + ASC) on zebrafish larvae because both concentrations did not cause any significant alteration in single exposure experiment. No significant change was observed in the general cardiac morphology after incubation in ASC + PON combination (Figure 4A–D). In addition, no significant alteration was observed in every cardiac performance parameter tested like stroke volume, cardiac output, shortening fraction, ejection fraction, heart rate, and heart rate variability (Figure 4E–N), suggesting no potential synergistic cardiac toxicity effect occurred following the co-incubation of ASC + PON.

### 2.4. Zebrafish Vascular Performance after Co-Incubation in PON and ASC

Since no significant cardiotoxicity was observed in zebrafish after the combination experiment, we further explored the potential vascular toxicity of the PON + ASC combination. When PON was administered alone on zebrafish, a significant decrease in blood flow velocity was observed starting from 1 ppm, and at 2 ppm, the blood stopped flowing (Figure 5A). Figure 5B shows the quantification data. This result is comparable to the cardiac performance parameter data described in Figure 2. Interestingly, no significant change in the blood cell morphology and blood clot formation was observed during the video recording, eliminating the possibility of interference in hematopoiesis caused by PON treatment. The video of the slow blood flow velocity of zebrafish larvae after PON treatment can be found in Appendix A.

Treatment with PON + ASC combination did not cause significant alteration in cardiac performance parameters in zebrafish (Figure 3); however, mild alteration of vascular performance was noted. For instance, the combination caused a decrease in the maximum and average blood flow velocity (Figure 6A–C). The combination treatment also did not cause blood clot formation in the whole dorsal aorta area of every zebrafish tested, indicating the absence of any thrombotic events. However, the PON + ASC combination can potentially decrease the blood flow velocity in zebrafish, thus warrants a follow-up study to evaluate the vascular toxicity of this combination.

### 2.5. Expression of the Cardiovascular Genes in Zebrafish after Co-Incubation in PON and ASC

The expression of genes related to cardiovascular development was observed to understand further the potential molecular mechanism of cardiovascular toxicity of PON and ASC in combination treatment. The qRT-PCR result showed a significant increase in the level of *nxk2.5* after PON + ASC treatment (Figure 7A). This should be given importance as the *nxk2.5* gene is responsible for heart development and morphogenesis. Other cardiac markers like *tbx5*, *amhc*, *vmhc*, and *myh6* levels showed no significant difference after either after PON, ASC, or PON + ASC treatment (Figure 7B–E). The combination treatment also did not cause a substantial change in the mRNA levels of several genes related to vascular or blood cell development; there was no significant difference in *fli1*, *vegfaa*, *vegfab*, *gata1*, *hbae1*, *hbbe1*, and *hbae2* mRNA levels before and after the combination treatment (Figure 7F–L).

### 2.6. Molecular Docking of ASC and PON to BCR-ABL

The molecular interaction of the PON and ASC was assessed in a normal and mutated form of BCR-ABL1 and BCR-ABL2 through the molecular docking technique (Appendix A). PON, which had binding energy of −8.7 kcal/mol, binds with the key amino acid residue of BCR-Abl1 (PDB ID: 5MO4) via Conventional Hydrogen Bond with Glu481, Carbon Hydrogen Bond with Ala356, Alkyl Bond with Leu359 and Tyr454, Pi-Alkyl with Val525, Pi-Sigma with Met456, Halogen (Fluorine) binding with Asn355, and an unknown unfavorable bump. On the other hand, ASC had a binding energy of −9.6 kcal/mol and interacted with the key amino acid residues of BCR-Abl1 via Conventional Hydrogen Bond with Arg351, Ala452, and Glu481, Carbon Hydrogen Bond with Leu360, Alkyl Bond with Ala363, Leu448, and Val 487, Pi-Alkyl with Pro484, Pi-Sulfur with Met456 and Cys483, Pi-Pi T-shaped with Tyr454, Halogen (Fluorine) bond with Leu359, and an unknown unfavorable bump. Appendix A shows the docked structures of PON and ASC on the active site of 5MO4 protein. For the mutated BCR-Abl1 protein (PDB ID: 4TWP), PON, which had binding energy of −8.2 kcal/mol, binds with the key amino acid residue through Carbon Hydrogen Bond with Tyr253, Alkyl Bonds with Ala269, Ile315, and Phe317, Pi-Alkyl Bonds with Leu248, Val256, and Met318, Pi-Sigma with Leu370, Halogen (Fluorine) Bond with Gln252, Asn368 and Asp381, and an unknown favorable bond. Meanwhile, ASC had binding energy of −8.1 kcal/mol and interacted with the key amino acid residues of the mutated Abl1 protein via Conventional Hydrogen Bond with Met318, Carbon Hydrogen Bond with Asp381, Alkyl Bond with Ala269, Pi-Alkyl with Val256, Pi-Pi T-shaped with Tyr253, Pi-Sigma with Leu248 and Leu370, Halogen (Fluorine) Bond with Phe317, and an unknown unfavorable bump. Appendix A shows the docked structure of PON and ASC on the active site of 4TWP.

In the case of Abl2 (PDB ID: 3HMI), PON garnered a binding energy of −11.2 kcal/mol. It interacted with the key amino acid residues via Carbon Hydrogen Bonds with Met364, His407, and Gly429, Alkyl Bond with Ala315, Pi-Alkyl Bonds with Val345 and Ala426, Pi-Pi T-shaped Bond with Tyr299, Pi-Sigma Bonds with Leu294 and Leu416, and Halogen (Fluorine) Bonds with Asp409 and Phe428. An unfavorable bump was also observed. Meanwhile, ASC garnered binding energy of −8.9 kcal/mol and interacted with numerous key amino acid residues such as Conventional Hydrogen Bonds with Met364, Asp409, and Arg413, Alkyl Bonds with Leu294 and Ala315, Pi-Alkyl Bond with Ala426, Pi-Pi T-shaped Bond with Tyr299, Pi-Sigma Bond with Leu416, Halogen (Fluorine) Bonds with Glu362 and Tyr363, and an unfavorable bump. Appendix A shows the docked structure of PON and ASC on the active site of 3HMI. Lastly, for the mutated form of Abl2 (PDB ID: 3KKI), PON bind with the key amino acid residues via Carbon Hydrogen Bonds with His67 and Gln104, Pi-Sigma Bond with Val63, Halogen (Fluorine) Bonds with Glu97, and Gln101, and an unfavorable bump. PON obtained binding energy of −7.9 kcal/mol. On the other hand, ASC interacted with numerous key amino acid residues such as Conventional Hydrogen Bonds Asp71 and Gln101, Carbon Hydrogen Bonds with His67, Ser74, and Glu97, Alkyl Bond with Leu100, Pi-Alkyl Bonds with Ala34 and Leu70, Halogen (Fluorine) Bond with Leu100, and an unfavorable bump. This compound had binding energy of −8.9 kcal/mol. Appendix A shows the docked structure of PON and ASC on the active site of 3KKI.

## 3. Discussion

The present study provides the detailed effect of acute exposure to PON and ASC and the potential impact of their combination on the cardiovascular physiology of zebrafish larvae for the first time. A significant decrease in a dose-dependent manner was observed in every cardiovascular performance parameter, e.g., stroke volume, cardiac output, ejection fraction, shortening fraction, heart rate, maximum blood flow velocity, and average blood flow velocity of zebrafish after PON exposure. Furthermore, a significant alteration was also observed in the cardiac rhythm, shown by a higher standard deviation of beat-to-beat timing and the difference in heart rate in the atrium and ventricle, which is the arrhythmia phenotype. Contrary to PON, ASC exposure did not cause any observable effect in every cardiac performance parameter tested, while cardiac rhythm was still comparable with the control group. A follow-up experiment that exposed the zebrafish to a combination (PON + ASC) was performed to check the possible synergistic cardiovascular toxicity of both compounds. Sub-effective concentrations of both compounds showed no significant alteration in every cardiac physiology test. However, zebrafish exposed to the combination of both compounds displayed a significant decrease in blood flow velocity. To further check the effect of combination treatment on the genes related to cardiovascular physiology, we used qPCR to check possible gene alterations. However, no significant alteration in mRNA level in the genes was observed, except *nxk2.5*, which significantly increased after exposure to both compounds.

Actually, cardiac toxicity of PON in zebrafish has been reported before. A previous study by Singh et al. reports that PON causes cardiac-specific abnormalities as shown by systolic ventricular defect and cardiomyocyte apoptosis in zebrafish larvae [33]. Another study by Cheng et al. demonstrated that PON exposure to zebrafish causes vascular defects, characterized by constriction in the dorsal aorta and posterior cardinal vein [34]. Similar to these findings, the present study also reported that even a 2.5 ppm of PON exposure caused a significant decrease in every cardiac physiology parameter, including the change in cardiac rhythm shown by arrhythmic heartbeat. Furthermore, a reduction in blood flow velocity was also observed starting from 1 ppm of PON treatment while at 2 ppm concentration, no blood circulation was observed. Some studies suggest that this might be caused by inhibition of phosphorylation of AKT at Thr308 and Ser473 sites which disturb PI3K/AKT signaling pathways [33]. Decrease phosphorylation of ERK1/2 was also reported after PON exposure, which suggests that PON exposure could lead to abnormal cardiac homeostasis and decrease cardiomyocyte survival [33,35,36,37,38]. However, from the blood flow velocity analysis, we saw that the oscillation pattern of zebrafish blood flow velocity was different after exposure to PON. At 1.5 ppm PON, empty gaps were observed for each rhythm, which is not observed in other concentrations. It is considered one of the downsides of this method because it can only calculate moving objects from frame to frame by using the stack difference plug-in in ImageJ. This plugin was essential to filter the unmoving background object, which screened some non-moving blood cells with the background noise. In the case of zebrafish with slow blood flow velocity, the blood cell stop for each oscillation; thus, the tool was unable to track the blood cell [39]. Nevertheless, we hypothesized that the observed toxicities of PON might be related to its broad spectrum of action since PON was designed to block the ATP binding site of tyrosine kinase, which is not specific to ABL1 only. The previous study showed that PON can block the activity of other kinases in other signaling pathways, including FGFR, VGFR, JAK/STAT, PDGFR, SRC, PI3K/AKT, ERK, and many more [11,35,37]. Thus, this broad spectrum of blockage was likely the reason for the high potency of PON toxicity. The present study includes the aspects of cardiovascular toxicity, which is often neglected as most research is focused on ABL protein. In addition, it is also intriguing to evaluate the expression levels of some important heart marker genes since in the present study; the lowest concentration of PON did not alter the expression level of those genes.

On the other hand, this study observed no significant difference in zebrafish cardiac physiological parameters after ASC exposure, even at the highest concentration tested. This result was in line with the recent analysis by Singh et al. that exposed zebrafish larvae for 72 h. The study reported no significant difference in ventricular shortening fraction after exposure to up to 10 µM of ASC [33]. Another study by Cheng et al. also reports a similar result that 48 h of exposure to up to 20 µM of ASC does not lead to noticeable cardiovascular toxicities in zebrafish larvae [34]. In contrast to PON which targets the ATP binding site of the kinase, ASC was designed to bind to the specific myristoyl site of ABL1 protein, which induces the formation of inactive kinase conformation [16]. In humans, ABL1 encodes a cytoplasmic and nuclear protein kinase that involves differentiation, division, and adhesion of cells while also regulating stress responses such as DNA damage [40,41]. In the future, this topic can be done to understand the possible toxicity caused by ASC entirely. In spite of that, the present study demonstrates that ASC is much less toxic than PON in terms of cardiovascular safety. Despite the observations on the safety and efficacy of ASC, several hematological events and adverse cardiovascular cases were associated with chronic exposure to ASC in CML patients [18,42].

Interestingly, no significant change was observed in the cardiac performance after acute incubation with combined ASC and PON. The result is surprising because PON was identified as the most cardiotoxic of all approved CML TKIs in zebrafish [16]. TKIs used in CML management, except for ASC, target the ATP binding pocket in the Abl1 kinase [11]. PON is a third-generation TKI with the broadest spectrum of its targets due to its triple bond ethynyl linker, which allows it to span the bulky residue side chains in the ATP-binding site [43]. The mechanism of action of ASC as an allosteric inhibitor is independent of the conformational change of the A loop, the target of ATP-competitive inhibitors [44]. While the value of combining ATP site and allosteric TKIs in the context of kinase inhibition has been investigated [16], our studies of combination treatment with ASC and PON in zebrafish demonstrate the capacity of such a combination to reduce cardiovascular toxicity associated with PON use. We hypothesized that this result is due to the higher binding affinities and more pronounced allosteric inhibitory effect of ASC that may have outcompeted the ATP-binding activity of PON. As reported, ASC binds explicitly to the myristate pocket rather than other orthosteric sites and has a particular on-target effect against the ABL1 kinase without any significant target effects [16,43,44]. Docking results show that PON and ASC bind on the surface of mutated BCR-ABL. The binding area possibly represents the ATP-binding site and myristoyl-binding pockets on the ABL kinase domain since the allosteric myristoyl-binding pocket is distant from the catalytic site in the kinase domain of BCR-ABL1 [33]. Whereas ASC has been postulated to be relatively specific for BCR-ABL1, PON reportedly binds to several additional target kinases [45], which possibly explains the overlapping of PON and ASC binding sites in the mutated ABL1 and ABL2 in the present docking studies. These findings suggest that PON and ASC bind with ABL1, ABL2, and mutated ABL1 and ABL2 proteins that are reported to mediate cardiovascular toxicity in CML patients. Furthermore, we also found a normal expression level of genes related to blood cell development, except *nxk2.5.* This gene is expressed during early cardiac morphogenesis and serves as a master regulatory protein. Because of its critical role in cardiogenesis, *nxk2.5* has been a prime candidate in studies to identify the genetic basis of structural congenital heart defects [46]. However, we speculated that the effects of its abnormal expression level could be compensated by other heart development genes since generally, most cardiovascular system development is coordinated by several specific transcription factors [47].

However, although the ASC and PON combination did not cause any significant cardiotoxicities, an abnormality in blood flow was still displayed by the larvae. To the best of the author’s knowledge, only a few studies focused on the side effect of the combination in animal and human studies. Previously, Zerbit et al. requested an early access program (EAP) of ASC and PON to treat a patient with acute lymphoblastic leukemia. However, no clinical and biological side effect was observed except persistent thrombocytopenia [48]. Several studies have reported that ASC and PON could synergistically inhibit various BCL-ABL mutants, and the combination can help reduce cardiovascular risk, mainly from PON usage. Clinical data have shown that the combination of PON + ASC was preferable for treating a patient with CML; for instance, adding 50 nM ASC can effectively reduce the IC_50_ of PON by 5.6-fold for BCR-ABL with T315M mutation and more than 7-fold for BCR-ABL with Y253/T315I mutation [44,45]. A previous study has demonstrated that mice treated with 30 mg/kg PON for 16 days did not manifest symptoms of hematological toxicity [49]. Mu et al. also reported a similar result when they administered 5 mg/kg of PON to mice for four weeks [50]. However, thrombocytopenia, neutropenia, and anemia were reported in phase 2 clinical trials involving PON suggesting that long-term exposure might cause hematotoxicity [51]. In this study, the combination treatment did not alter the mRNA level of the genes related to zebrafish blood cells and vascular development. However, typical side effects (>10%) associated with hematological toxicity, such as decreased hemoglobin, neutrophils, platelet counts, and lymphocytopenia, were observed in patients at the recently concluded clinical trial of ASC [42]. Based on these findings, we hypothesized that the combination of these compounds in the current concentrations was not enough to diminish the toxicities of PON, leaving the observed blood flow abnormalities in the treated group as also displayed in the PON-treated group. In addition, further experiments are warranted to carefully investigate whether the combinations of PON and ASC in chronic uses would produce unexpected adverse cardiovascular effects on zebrafish.

Based on the current and previous studies, it is safe to say that ASC is safer compared to PON in case of a cardiovascular event. Previously, through the development of TKIs, PON was regarded as the most effective TKI compared to the other due to its wide range of target and its effectiveness against the resistance T315I mutant [11]. However, due to the severe cardiovascular side effect, it was placed as a last resort when no other TKI was available to help patients with leukemia. ASC, which was a newly developed TKI, is the only TKI that targets the myristoyl pocket of BCR-ABL1 protein. However, similar to other TKIs, some mutants also develop resistance to ASC like A3437, P465S, and V468F mutant [44]. Thus, the potential of a combination of ASC and another TKI like PON is a good approach to treating a patient with leukemia as it showed synergistic interaction in silico [44]. Although right now, in some emergency cases, a combination of ASC and PON is already been used for patients with leukemia [48], the potential side effect after the admission of both compound is not been fully explored yet. Thus, the success of this research by using zebrafish animal models in studying the cardiovascular toxicity of targeted therapy in CML will help advance our basic research achievement in this field in Taiwan. The long-term goal of this study is to open more research to understand the mechanisms by which targeted therapy in CML causes cardiovascular toxicity and to reduce or prevent cardiovascular toxicity in patients with CML.

## 4. Materials and Methods

### 4.1. Animal Ethics and Drug Exposure

All tests including zebrafish were performed following the rules affirmed by the Institutional Animal Care and Use Committees (IACUCs) of the Chung Yuan Christian University. Wild-type AB strain zebrafish were utilized in this study as a vertebrate animal model and kept up in a ceaselessly sifted and circulated air through a water framework. The temperature was kept up at 26 ± 1 °C with 14/10 h of light/dark cycle agreeing to already detailed protocols. For egg collection, male and female zebrafish with a proportion of 2:1 were moved to the breading chamber one night before. The next morning, the separator was removed, and the eggs were collected in two hours. After the eggs were collected, they were moved into an incubator with a temperature of 28 ± 1 °C until the treatment time.

PON and ASC were purchased from Aladdin Chemicals (Shanghai, China). The compounds were diluted with DMSO to make a stock solution and kept at −20 °C and 4 °C for PON and ASC, respectively. The stock concentration was further diluted with ddH_2_O into desired concentration at the time of testing. At 48 hpf, zebrafish larvae were exposed to PON and ASC for 24 h, and at 72 hpf, the cardiovascular system was assessed. The experiments were done in triplicate with an average of seven individuals per compound for each replication.

### 4.2. Acute Toxicity Assay

The acute toxicity assay was performed according to the OECD 203 Guidelines. About 12 healthy zebrafish eggs were selected at 24 hpf and exposed to both compounds for 96 h. The dead zebrafish were removed each day, and the mortality rate was observed at the end of the experiment. The lethal concentration 50 (LC_50_) was calculated and used as a basis for concentration selection.

### 4.3. Cardiovascular Performance Measurement

In recording zebrafish heartbeats and cardiac physiology parameters, methylcellulose with a 3% concentration was used as a mounting agent during video recording to minimize zebrafish movement. A high-speed digital charged coupling device (CCD) (AZ Instruments, Taichung, Taiwan) was mounted on an inverted microscope (Sunny Optical Technology, Yuyao, China) to record zebrafish heartbeat and blood flow. For better video clarity, Hoffmann objective lenses with 40× magnification were used. The video was recorded for 10 s at 200 frames per second (fps) following the previously published protocol [52]. The “Trackmate” plug-in ImageJ software was used to calculate the blood flow velocity of zebrafish. At the same time, heart rate analysis was performed using the Time Series Analyzer V3 plug-in (https://imagej.nih.gov/ij/plugins/time-series.html, accessed on 20 January 2022) to analyze the change of dynamic pixel intensity and get the pattern of systolic and diastolic from frame to frame. Heart rate was expressed as beats per minute (bpm) and measured using Peak analyzer function in OriginPro 2019 software (Originlab Corporation, Northampton, MA, USA) by determining the time interval of each peak. To calculate the heartbeat variability, the sd1 (standard deviation 1) and sd2 (standard deviation 2) extracted from the Poincare plot plug-in from the OriginPro 2019 software were recorded and statistically analyzed. Sd1 refers to the immediate beat-to-beat variability of the heart which can be caused by parasympathetic modulation while sd2 refers to the long-term variability of the heartbeat which reflects sympathetic activation. For example, a patient with diabetes mellitus will have a lower sd1 value due to weakened parasympathetic regulation and a higher sd2 value due to compensatory sympathetic input [53]. Stroke volume was determined by the assumption that the heart chamber has an ellipsoid shape and was calculated by subtracting end-systolic volume (ESV) from the end-diastolic volume (EDV). The heart chamber volume was calculated using the heart chamber long (DL) and short axis (DS) during EDV and ESV. Cardiac output was calculated by multiplying the heart rate observed in the ventricle with stroke volume. The ejection fraction and shortening fraction were important parameters that represent heart’s muscular contractility and calculated using the following formula:$$EF\left(\%\right)=\frac{SV}{EDV}\times 100\%$$
$$SF\left(\%\right)=\frac{Ds\left(EDV\right)-Ds\left(ESV\right)}{Ds\left(EDV\right)}\times 100\%$$

### 4.4. Biomarker Expression Assay to Elucidate Potential Toxicity Mechanism

Live embryos were collected, and several gene expressions related to cardiac development were selected, including several transcription factors *myc6*, *tbx5*, *nkx2.5*, and the myosin heavy chains *amhc* and *vmhc*. Gene expression of *vegfaa* and *vegfab* was also used to check the vascular development of zebrafish larvae. Furthermore, some biomarkers related to red blood cell oxygen binding and development were also analyzed to explore the mechanism of toxicity. qRT-PCR was performed using a commercial kit following the manufacturer’s instructions. The experiment had three replicates per sample (*n* = 30/sample). The sequence of each primer used for qRT-PCR is summarized in Appendix A.

### 4.5. Molecular Docking of Asciminib and Ponatinib to BCR-ABL

The molecular interaction of PON and ASC with normal and mutated BCR-ABL genes was virtually examined through the molecular docking process using Autodock Vina 1.1.2 in Python Prescription (PyRx) (https://autodock-vina.readthedocs.io/en/latest/docking_python.html accessed on 20 January 2022). The crystal structures of BCR-ABL1 (PDB ID: 5MO4), Mutated BCR-ABL1 (PDB ID: 4TWP), BCR-ABL2 (PDB ID: 3HMI), and Mutated BCR-ABL2 (PDB ID: 2KK1) were downloaded from Research Collaboratory for Structural Bioinformatics (RCSB) Protein Data Bank. The ligands in their three-dimensional (3D) forms were obtained from the PubChem compound database. The binding modes, energy, and residue interaction of PON and ASC on the selected active site were determined. The grid optimization process was performed using Autodock Vina, and the grid box parameters are shown in Appendix A and enclose the surface of the amino acid residues of the target binding site.

### 4.6. Biostatistics

Statistical analysis was done using GraphPad Prism (GraphPad Inc., La Jolla, CA, USA). The normality and relative standard deviation measurements were initially performed to select the appropriate statistical test. Based on the normality of the data distribution of the sample ANOVA tests were performed accordingly. The data’s variance and significance were calculated based on the appropriate post-hoc multiple comparison test.

## Figures and Tables

**Figure 1 ijms-23-11711-f001:**
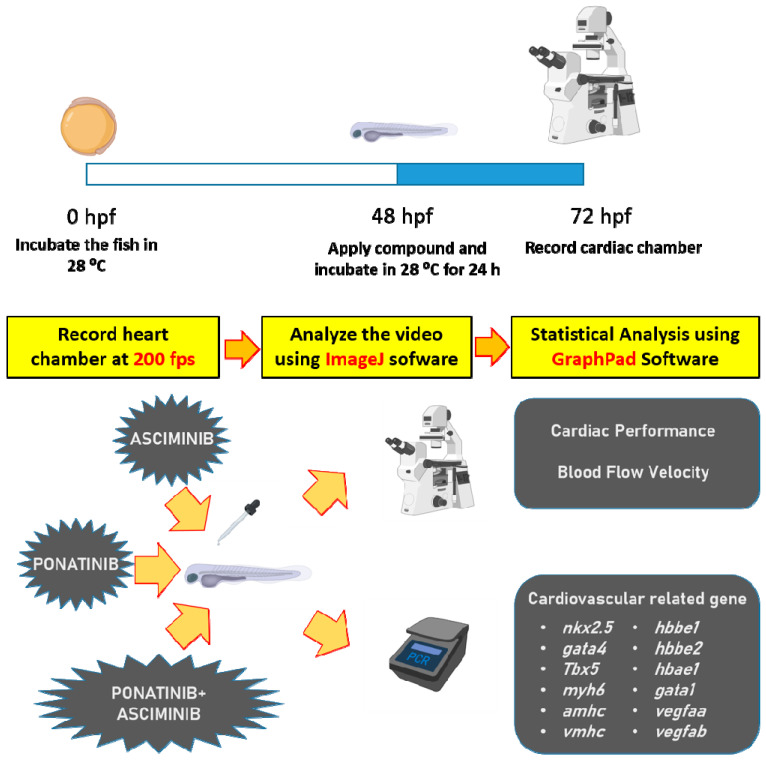
Schematic diagram showing the overall experimental design for the study. The upper panel shows the timing and workflow to conduct cardiovascular assessments in zebrafish. The bottom panel shows the physiological and molecular endpoints used to evaluate the potential adverse effect of the tested chemicals.

**Figure 2 ijms-23-11711-f002:**
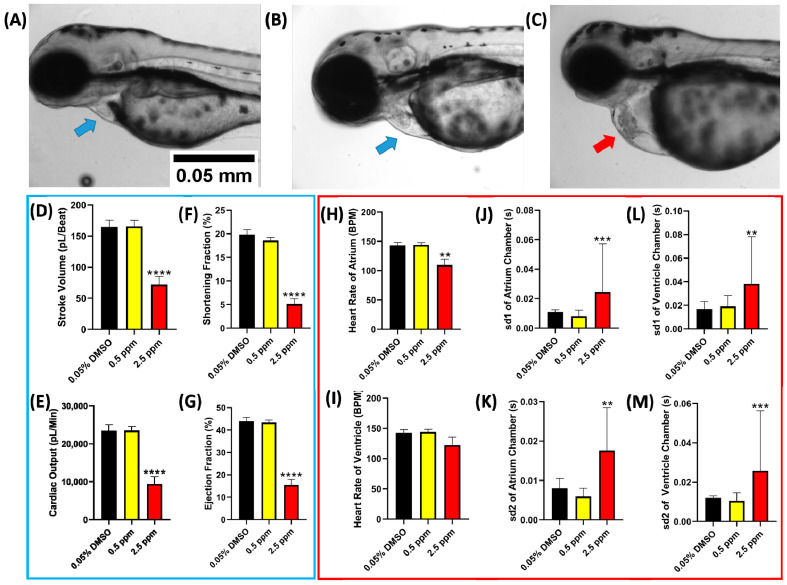
The morphology of 72 hpf zebrafish larvae after acute exposure in 0 (**A**), 0.5 (**B**), and 2.5 ppm (**C**) of PON (blue arrow show normally developed cardiac chamber while red arrow show abnormal development of cardiac chamber). Zebrafish cardiac physiology parameters (**D**–**G**, blue box) and cardiac rhythm parameters ((**H**–**M**), red box) after exposure to PON. Data are presented as mean ± standard deviation, and the statistical significance was calculated using ordinary one-way ANOVA with Dunnet multiple comparison test. (** *p* < 0.01, *** *p* < 0.001, **** *p* < 0.0001).

**Figure 3 ijms-23-11711-f003:**
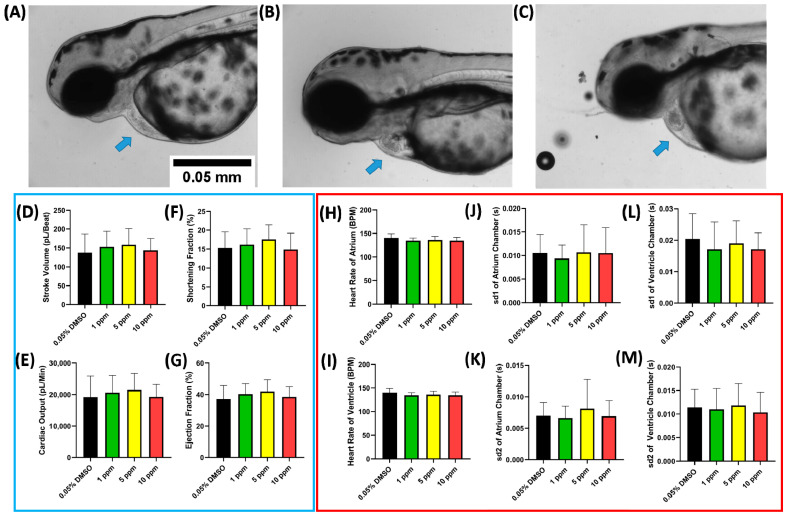
The morphology of 72 hpf zebrafish larvae after acute exposure in 0 (**A**), 5 (**B**), and 10 ppm (**C**) of ASC (blue arrow shows the normally developed cardiac chamber). Zebrafish cardiac physiology parameters ((**D**–**G**), blue box) and cardiac rhythm parameters ((**H**–**M**), red box) after exposure to ASC. Data are expressed as mean ± standard deviation, and the statistical significance was calculated using ordinary one-way ANOVA with Dunnet multiple comparison test (degree of significance was set to *p* < 0.05).

**Figure 4 ijms-23-11711-f004:**
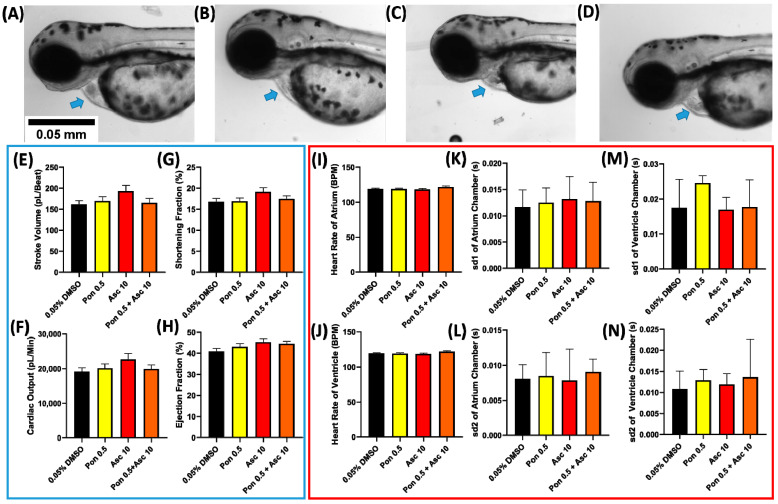
The morphology of 72 hpf zebrafish larvae after acute exposure in 0 (**A**), 0.5 ppm of PON (**B**), 10 ppm of ASC (**C**), and a combination of 0.5 and 10 ppm of PON and ASC (**D**) (blue arrow show the normally developed cardiac chamber). Zebrafish cardiac physiology parameter ((**E**–**H**), blue box) and cardiac rhythm parameter ((**I**–**N**), red box) after exposure to PON and ASC. Data are expressed as mean ± standard deviation, and the statistical significance was calculated using ordinary one-way ANOVA with Dunnet multiple comparison test (degree of significance was set to *p* < 0.05).

**Figure 5 ijms-23-11711-f005:**
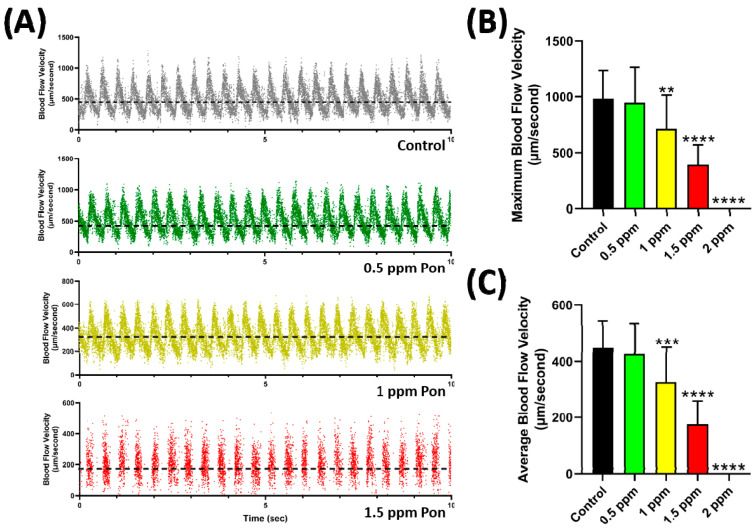
Representative of zebrafish larvae blood flow oscillation pattern after incubation in 0.05% DMSO (grey), 0.5 (green), 1 (yellow), and 1.5 (red) ppm of PON (dash line show the average velocity) (**A**). Maximum (**B**) and average (**C**) blood flow velocity in the dorsal aorta of zebrafish larvae after exposure to PON. Data are presented as mean ± standard deviation, and the statistical significance was calculated using ordinary one-way ANOVA with Dunnet multiple comparison test. (** *p* < 0.01, *** *p* < 0.001, **** *p* < 0.0001).

**Figure 6 ijms-23-11711-f006:**
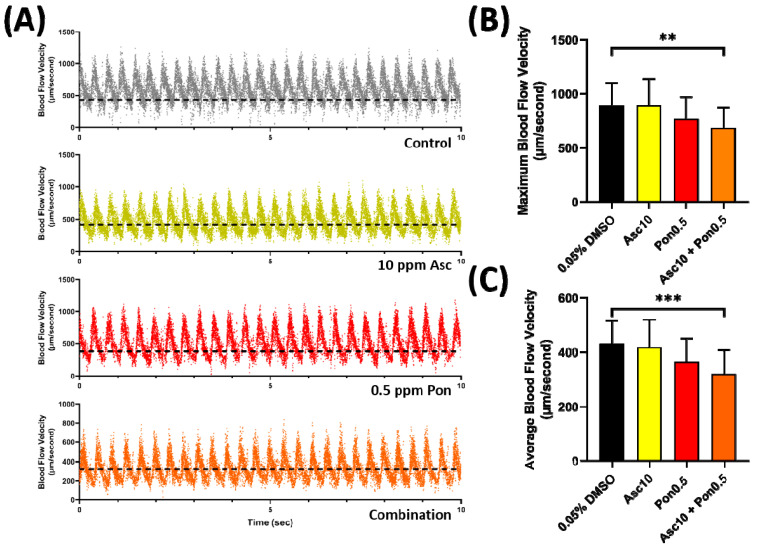
Representative zebrafish larvae blood flow oscillation pattern after incubation in 0.05% DMSO (grey), 10 ppm of ASC (yellow), 0.5 ppm of PON (red), and a combination of both compound (orange) (dash line show the average velocity) (**A**). Maximum (**B**) and average (**C**) blood flow velocity in the dorsal aorta of zebrafish larvae after exposure to ASC and PON. Data are presented as mean ± standard deviation, and the statistical significance was calculated using ordinary one-way ANOVA with Dunnet multiple comparison test. (** *p* < 0.01, *** *p* < 0.001).

**Figure 7 ijms-23-11711-f007:**
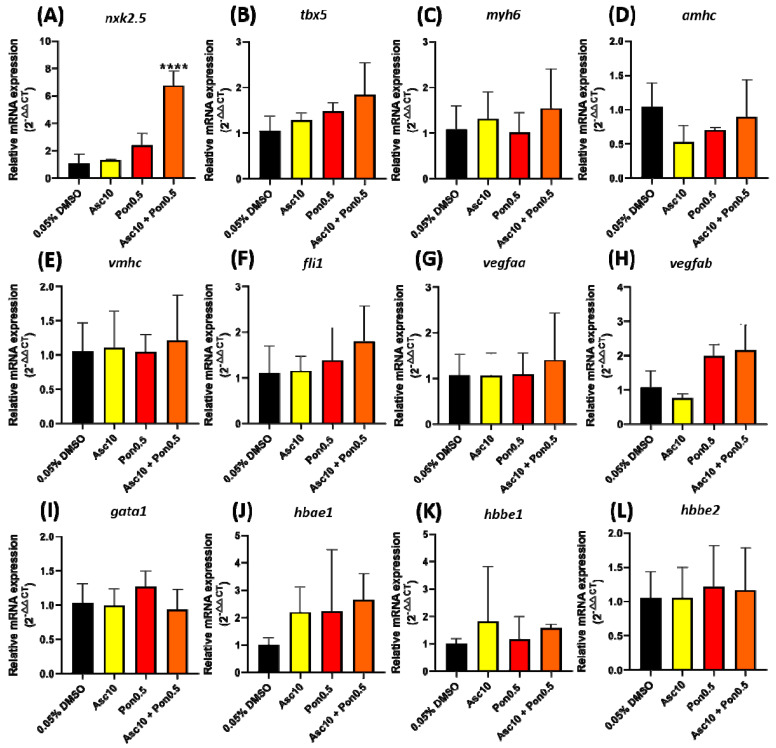
Relative mRNA level of cardiovascular-related genes in zebrafish after exposure to ASC, PON, and combining both compounds. Some molecular makers associated with cardiac development (**A**–**E**), vascular development (**F**–**H**), and red blood cell development (**I**–**L**) were tested. The value was expressed as mean ± standard deviation, and the statistical difference was calculated using ordinary one-way ANOVA with Dunnet multiple comparison test. (**** *p* < 0.0001).

## Data Availability

The datasets used and/or analyzed during the current study are available from the corresponding author upon reasonable request.

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
