# Peer review of "Investigating Potential Cardiovascular Toxicity of Two Anti-Leukemia Drugs of Asciminib and Ponatinib in Zebrafish Embryos"

_ijms, 2022, doi:10.3390/ijms231911711_

Round 1

Reviewer 1 Report

This is an interesting manuscript that describes the use of zebrafish (ZF) embryos to study the cardiovascular effects of two tyrosine kinase inhibitor (TKIs).  The TKIs, Asciminib (ASC) and Ponatinib (PON), are on the cutting edge of treatment for CML.   The topic is clinically meaningful and this manuscript would be a worthy contribution to paving the way for TKIs to be safely introduced into clinical care.

   There are however a number of issues that substantially detract from the readability and impact of the manuscript.  These issues are discussed in greater detail below:

  1. Figure 1 design issue.   The color choices of the bottom portion of Figure 1 is very unfortunate and should be changed.   The text in white font in against a light blue background in the starry ovals and squares with rounded corners are exceedingly and unnecessarily difficult to read.  Please change to high contrast, either white text against a DARK background, or Black/Dark text against a very light or white background.

  2. Lack of Positive controls.  The experiments all lack a positive control.  Can the investigators show at least one experiment in which ZF embryos are exposed to an agent with known cardiovascular toxicity (besides ASC or PON, such as adriamycin?), to show what the altered morphology looks like?   And also the effects of the positive control on cardiac physiology and rhythm parameters?   The ZF morphology alterations after the exposure to the highest concentrations of PON and ASC shown in Figures 2A and 3A should be indicated with an arrow or triangle that points to that portion of the anatomy affected (ideally also pointed out in a picture of the positive control).   Finally, for Figure 2D-M, it would be ideal to have a histogram that shows the effects of both a negative control (water only?) on cardiac physiology and rhythm parameters? 

   3.   SD1 and SD2 parameters.  It would be helpful to have an explanation in the Materials and Methods of what the sd1 and sd2 parameters are --no explanation is currently provided in the manuscript.   I had to Google to find the following on the internet:

  "Poincarè plot analysis SD1 (ms) Indicator of the standard deviation of the immediate RR variability due to parasympathetic efferent (vagal) influence on the sino-atrial node

SD2 (ms) Indicator of the standard deviation of the slow variability of the heart rate. It is accepted that this value is representative of the global variation in HRV"

    Is this what how sd1 and sd2 as utilized in this manuscript is defined?  Is "sd "an abbreviation of "standard deviation" or "standard determinants"?It should also be briefly explained the clinical significance of RR and HRV, as sd1 and sd2 relates to these.    

   4. Figure 2 and 3 design.   The layout and color choices of the histograms in Figure 2D-M and Figure 3D-M are suboptimal and should be changed.  The choice of Black color is used for 0.05% DMSO in Figure 2 but for Water only in Figure 3 --this is very confusing and distracting.   As mentioned above, Figure 2 should also have a water only control, and its histogram colored the same as the water only histogram for Figure 3.   Why should Figure 2 have three bars, and Figure 3 have four?    The D-G figures should have a box around this group, to demarcate the cardic physiology parameters, while H-M should be outlined with another box to demarcate the cardiac rhythm parameters.   

   5. Choice of PON, ASC or PON + ASC.  There should be more detailed Discussion about how current or future standard of care would incorporate PON or ASC?   Since ASC has lower risk of toxicity, would this be increasingly preferred over PON?   Or for some patients, PON would still be added to ASC, e.g. for disease resistant to or recurrent after ASC?

Author Response

Comments and Suggestions for Authors

This is an interesting manuscript that describes the use of zebrafish (ZF) embryos to study the cardiovascular effects of two tyrosine kinase inhibitor (TKIs). The TKIs, Asciminib (ASC) and Ponatinib (PON), are on the cutting edge of treatment for CML. The topic is clinically meaningful and this manuscript would be a worthy contribution to paving the way for TKIs to be safely introduced into clinical care.

There are however a number of issues that substantially detract from the readability and impact of the manuscript. These issues are discussed in greater detail below:

  1. Figure 1 design issue. The color choices of the bottom portion of Figure 1 is very unfortunate and should be changed. The text in white font in against a light blue background in the starry ovals and squares with rounded corners are exceedingly and unnecessarily difficult to read. Please change to high contrast, either white text against a DARK background, or Black/Dark text against a very light or white background.

Thank you for the constructive suggestion. The authors were aware of the issue regarding the lack of visibility of the text in the starry ovals and squares parts in Figure 1. Therefore, the authors agreed with the reviewer’s suggestion since it could cause difficulties to read. Thus, the background of the mentioned parts had been changed to a darker color while the text was changed to brighter color and bolder and clearer font to provide a high contrast so the readers can read and understand the figure easily.

  1. Lack of Positive controls. The experiments all lack a positive control. Can the investigators show at least one experiment in which ZF embryos are exposed to an agent with known cardiovascular toxicity (besides ASC or PON, such as adriamycin?), to show what the altered morphology looks like? And also the effects of the positive control on cardiac physiology and rhythm parameters? 

The authors appreciated the detailed questions. It is true that besides the solvent control group (0.05% DMSO), no data from other control groups were provided in the current study. Although positive control might be important as the reviewer suggested, this decision was taken after considering several factors. First, as mentioned in the manuscript, several prior studies of PON had confirmed that it could cause significant adverse cardiovascular-related events, thus, supporting the current claim of its cardiotoxicity (Chen et al., 2021; Singh et al., 2019; Suryanto et al., 2022; Zhu et al., 2020). Next, regarding the comparison of the PON effects with another known agent that is known to possess cardiovascular toxicities, the previous findings had demonstrated that ponatinib could significantly reduce cardiac performance together with cardiac defects and edema in zebrafish larvae which have comparable results with exposure to high concentrations of ethanol (Li et al., 2016; Suryanto et al., 2022). Therefore, by including this reference in the manuscript, the authors believe that it can help readers to understand the cardiotoxicity level of PON shown in the present study as they can compare it with the effects of a known cardiotoxicity agent. Finally, this decision was also taken after considering the animal usage reduction since here, the sample number used was already relatively high because of the number of the groups applied and also to provide high-reliability data.  

Chen, W., Xie, L., Yu, F., Li, Y., Chen, C., Xie, W., Huang, T., Zhang, Y., Zhang, S., & Li, P. (2021). Zebrafish as a model for in-depth mechanistic study for stroke. Translational Stroke Research, 12(5), 695-710.

Li, X., Gao, A., Wang, Y., Chen, M., Peng, J., Yan, H., Zhao, X., Feng, X., & Chen, D. (2016). Alcohol exposure leads to unrecoverable cardiovascular defects along with edema and motor function changes in developing zebrafish larvae. Biology open, 5(8), 1128-1133.

Singh, A. P., Glennon, M. S., Umbarkar, P., Gupte, M., Galindo, C. L., Zhang, Q., Force, T., Becker, J. R., & Lal, H. (2019). Ponatinib-induced cardiotoxicity: delineating the signalling mechanisms and potential rescue strategies. Cardiovascular research, 115(5), 966-977.

Suryanto, M. E., Saputra, F., Kurnia, K. A., Vasquez, R. D., Roldan, M. J. M., Chen, K. H.-C., Huang, J.-C., & Hsiao, C.-D. (2022). Using DeepLabCut as a Real-Time and Markerless Tool for Cardiac Physiology Assessment in Zebrafish. Biology, 11(8), 1243.

Zhu, X.-Y., Xia, B., Ye, T., Dai, M.-Z., Yang, H., Li, C.-Q., & Li, P. (2020). Ponatinib-induced ischemic stroke in larval zebrafish for drug screening. European Journal of Pharmacology, 889, 173292.

The ZF morphology alterations after the exposure to the highest concentrations of PON and ASC shown in Figures 2A and 3A should be indicated with an arrow or triangle that points to that portion of the anatomy affected (ideally also pointed out in a picture of the positive control). 

Thank you for the comment. The authors strongly agreed with the reviewer's suggestion in adding an arrow to indicate the morphology alterations in zebrafish larvae after exposure of PON and ASC. Therefore, an arrow that points to that portion of the anatomy affected was added in each figure 2A-C, 3A-C, and 4A-C as the reviewer suggested.

Finally, for Figure 2D-M, it would be ideal to have a histogram that shows the effects of both a negative control (water only?) on cardiac physiology and rhythm parameters? 

The authors appreciated the questions. As mentioned above, the authors believed that the current control (0.05% DMSO) was sufficient enough to show the cardiotoxicities of PON. This decision was taken because several previous studies demonstrated that DMSO in this concentration has only a minor to no effect on zebrafish larvae (Hallare et al., 2004; Hallare et al., 2006; Xiong et al., 2017). Therefore, the authors presumed that a negative control group that is treated with water only would have comparable results with the current DMSO group. In addition, similar to the positive control, this negative control was also not provided to reduce the number of used animals in the present study.

Hallare, A., Köhler, H.-R., & Triebskorn, R. (2004). Developmental toxicity and stress protein responses in zebrafish embryos after exposure to diclofenac and its solvent, DMSO. Chemosphere, 56(7), 659-666.

Hallare, A., Nagel, K., Köhler, H.-R., & Triebskorn, R. (2006). Comparative embryotoxicity and proteotoxicity of three carrier solvents to zebrafish (Danio rerio) embryos. Ecotoxicology and environmental safety, 63(3), 378-388.

Xiong, X., Luo, S., Wu, B., & Wang, J. (2017). Comparative developmental toxicity and stress protein responses of dimethyl sulfoxide to rare minnow and zebrafish embryos/larvae. Zebrafish, 14(1), 60-68.

  1. SD1 and SD2 parameters. It would be helpful to have an explanation in the Materials and Methods of what the sd1 and sd2 parameters are --no explanation is currently provided in the manuscript. I had to Google to find the following on the internet:

"Poincarè plot analysis SD1 (ms) Indicator of the standard deviation of the immediate RR variability due to parasympathetic efferent (vagal) influence on the sino-atrial node

SD2 (ms) Indicator of the standard deviation of the slow variability of the heart rate. It is accepted that this value is representative of the global variation in HRV"

Is this what how sd1 and sd2 as utilized in this manuscript is defined?  Is "sd "an abbreviation of "standard deviation" or "standard determinants"? It should also be briefly explained the clinical significance of RR and HRV, as sd1 and sd2 relates to these.    

Thank you for the detailed questions. The authors understood the importance to include an explanation of several uncommon terms used in the manuscript. First, “sd” is an abbreviation of “standard deviation” and its value was obtained from the analysis using the Poincare Plot plug-in by OriginLab software. Next, the usage of sd1 and sd2 in the manuscript is to show the immediate RR variability and the slow variability of the heart rate, as the reviewer had mentioned above. Considering that the absence of these explanations might cause some difficulties to readers in understanding the results, the explanations of sd1 and sd2 and their relation with RR and HRV were added in the materials and methods section.

  1. Figure 2 and 3 design. The layout and color choices of the histograms in Figure 2D-M and Figure 3D-M are suboptimal and should be changed. The choice of Black color is used for 0.05% DMSO in Figure 2 but for Water only in Figure 3 --this is very confusing and distracting. As mentioned above, Figure 2 should also have a water only control, and its histogram colored the same as the water only histogram for Figure 3.

The authors appreciated the correction. Actually, there was a mistake regarding the group’s name in Figure 3 that might confuse readers. As mentioned above, the only control group used in the present study was the 0.05% DMSO group, which was the used solvent to dissolve both compounds, after several considerations mentioned above. Therefore, the “Control” group that was shown in the previous version manuscript referred to this group instead of a water-only group. The authors were aware of this inconsistency, thus, the “Control” group in Figure 3 was changed to “0.05% DMSO” to avoid confusion.

Why should Figure 2 have three bars, and Figure 3 have four? 

Thank you for the question. The difference in the number of bars between Figure 2 and Figure 3 is caused by the difference in the number of tested concentrations between PON and ASC. While three concentrations were tested in the ASC experiment (1, 5, and 10 ppm), only two concentrations were chosen in the PON study (0.5 and 2.5 ppm). As already mentioned in the manuscript, initially, several concentrations of PON and ASC were chosen in evaluating the potential acute cardiovascular toxicity. However, starting from the second highest concentrations of PON, which were 5 and 10 ppm, the mortality of zebrafish larvae was very high. Therefore, the authors decided to not include the data in the histograms since it will cause some problems during either cardiac rhythm or statistic calculations.

The D-G figures should have a box around this group, to demarcate the cardic physiology parameters, while H-M should be outlined with another box to demarcate the cardiac rhythm parameters.   

The authors thanked the reviewer for the constructive suggestion. It is true that in the previous version, it is quite difficult for the readers to distinguish which one is cardiac physiology or cardiac rhythm parameters. Thus, to help the readers in differentiating those parameters, a box around those groups was added with a blue color box for cardiac physiology parameters and a red color box for cardiac rhythm parameters according to the reviewer’s suggestion.

  1. Choice of PON, ASC or PON + ASC. There should be more detailed Discussion about how current or future standard of care would incorporate PON or ASC? Since ASC has lower risk of toxicity, would this be increasingly preferred over PON?   Or for some patients, PON would still be added to ASC, e.g. for disease resistant to or recurrent after ASC?

Thank you for the comment. The authors also felt the lack of discussion related to the choice between PON and ASC in the future based on the current findings. Therefore, more discussion regarding this matter was added to the Discussion section. Even though it is still case-by-case, based on the current results and previous findings, the usage of ASC is more preferred than PON for leukemia treatment if it passes clinical trial and approved by FDA to be used as a medicine in the future. However, PON still can be used if no other TKIs are available for treatment. In addition, its combination with ASC is also promising to be one of the decent approaches for treating a patient with leukemia in some emergency cases.

Reviewer 2 Report

Dear authors,

     This is an interesting study about potential cardiovascular toxicity of two anti-leukemia drugs of asciminib and ponatinib using Zebrafish embryos as model. The manuscript is comprehensive, well explained and with many figures illustrating the results.

   I have only one recommendation, it will be nice to have more explanation in the Introduction about Zebrafish models, even on other type of studies to show and to prove their importance (with citations), not only to say that is a "well-known animal model".

Best regards,

Author Response

Comments and Suggestions for Authors

Dear authors,

This is an interesting study about potential cardiovascular toxicity of two anti-leukemia drugs of asciminib and ponatinib using Zebrafish embryos as model. The manuscript is comprehensive, well explained and with many figures illustrating the results. I have only one recommendation, it will be nice to have more explanation in the Introduction about Zebrafish models, even on other type of studies to show and to prove their importance (with citations), not only to say that is a "well-known animal model".

Thank you for the constructive suggestion. The authors agreed that the previous version of the manuscript lacked an explanation regarding the zebrafish model. Therefore, some discussions and references were added to the manuscript as the reviewer suggested, highlighting the high usefulness of this aquatic vertebrate model since zebrafish have been used in many fields, including neuroscience, genetic manipulation, and aquatic toxicology, including water pollutant screening.

Round 2

Reviewer 1 Report

The authors have revised their manuscript to my satisfaction.   These contribute to a much improved manuscript and I have no further suggestions.